# Comparing Ionic Profile of Gingival Crevicular Fluid and Saliva as Distinctive Signature of Severe Periodontitis

**DOI:** 10.3390/biomedicines10030687

**Published:** 2022-03-17

**Authors:** Federica Romano, Giovanni Iaderosa, Matteo Corana, Stefano Perotto, Giacomo Baima, Federica Di Scipio, Giuliana Abbadessa, Giulia Maria Mariani, Mario Aimetti, Giovanni Nicolao Berta

**Affiliations:** 1Department of Surgical Sciences, C.I.R. Dental School, Section of Periodontology, University of Turin, 10126 Turin, Italy; federica.romano@unito.it (F.R.); giovanni.iaderosa@edu.unito.it (G.I.); matteo.corana@edu.unito.it (M.C.); stefanoperotto@libero.it (S.P.); giacomo.baima@unito.it (G.B.); giuliamaria.mariani@unito.it (G.M.M.); 2Department of Clinical and Biological Sciences, University of Turin, 10043 Orbassano, Italy; federica.discipio@unito.it (F.D.S.); giuliana.abbadessa@unito.it (G.A.)

**Keywords:** gingival crevicular fluid, ICP-MS, ICP-OES, ionomics, metallomics, periodontitis

## Abstract

Although increasing evidence is emerging on the contribution of chemical elements in periodontal health, no studies have concomitantly evaluated the ionic profile in gingival crevicular fluid (GCF) and saliva in relation to the underlying periodontal status. Our hypothesis is that these biofluids have distinctive ionic content. Therefore, the aim of this cross-sectional study was to analyze the elemental composition of GCF and saliva in order to explore which biological matrix and which combination of elements could discriminate between periodontitis and periodontal health. Twelve ions were analyzed in GCF and unstimulated saliva from 54 subjects (18 periodontally healthy, 18 untreated severe periodontitis and 18 treated severe periodontitis) using inductively coupled plasma–mass spectrometry (ICP-MS) and inductively coupled plasma–optical emission spectroscopy (ICP-OES). These analytical techniques were able to determine levels of sodium (Na), potassium (K), calcium (Ca) and magnesium (Mg), while the other elements were below the detection threshold. Na and K ions were detected at elevated concentration in untreated periodontitis compared with treated periodontitis and healthy periodontium. Ca was increased in untreated periodontitis, but the difference was not significant. In saliva, only Na was significantly associated with periodontitis. The combination of Na and K in GCF enabled the correct assignment of a subject to the periodontitis or healthy group. Based on these preliminary results, GCF demonstrated higher clustering potential than saliva.

## 1. Introduction

Periodontitis is a biofilm-induced chronic inflammatory disease that leads to the oxidative damage of soft and hard tooth-supporting tissues in susceptible individuals [1]. The interplay between microbial species and periodontal tissues is marked by the release in the oral fluids of tissue breakdown products, inflammatory mediators, bacterial by-products and metabolites [2]. Among them, increasing evidence supports the essential role of inorganic ions in regulating periodontal tissues homeostasis and in modulating immune-inflammatory and oxidative stress pathways [3]. Thus, their imbalance in the oral fluids may be reflective of the ongoing periodontal destructive process or it may be the consequence of the inflammatory milieu [4]. In this regard, ionomics is a promising approach to identify reliable markers for diagnosis, prognosis and clinical monitoring of periodontal tissue breakdown with saliva and gingival crevicular fluid (GCF) being the most attractive biological media [5]. Their collection is easy, non-invasive and minimal-time consuming.

Most of the currently available studies assessed the levels of metal ions in unstimulated whole saliva. In particular, major elements such as sodium (Na) and potassium (K) and trace elements such as calcium (Ca) and selenium (Se) were found at higher concentrations in saliva of chronic or aggressive periodontitis compared to gingivitis and periodontally healthy controls [6,7,8,9]. Furthermore, other studies reported a significant association between periodontitis and high salivary levels of copper (Cu), magnesium (Mg) and manganese (Mn) [10,11]. The exact role of each element in periodontal disease still remains to be fully elucidated. Regardless, it has been suggested that Cu and Mn are involved in the regulation of immune-inflammatory pathways [3], while Na and K in the preservation of the structural integrity of epithelial and connective tissues [12]. In addition, Mg and Ca modulate the alveolar bone remodeling mechanisms [13].

A recent paper from our research group compared the elemental composition of saliva in severe untreated periodontitis and periodontal health, including both intact and reduced periodontium after the completion of active periodontal therapy. Out of 10 target ions, Cu, Na, iron (Fe) and Mn appeared to be strictly associated with severe periodontitis. Lithium (Li) and zinc (Zn) were the only metal ions that were found to be significantly different between treated periodontitis subjects with restored healthy periodontium and healthy controls [14], contributing to the wound healing process [15].

Although GCF is the main source of biomarkers for periodontal disease, only few and dated studies linked GCF mineral content with periodontal status providing inconsistent results [12,16,17,18]. To the best of our knowledge, there are no available studies that evaluated concomitantly the ionic profile both in GCF and saliva. Our hypothesis is that these biofluids have distinctive elemental composition considering that saliva collects components from many other sources than GCF and mirrors the whole mouth inflammatory conditions. GCF is more influenced by the site-specific characteristics of the subgingival environment as fluid lying in close proximity to the periodontal tissues. Thus, this cross-sectional study was designed to extend our previous findings on salivary concentration of metal ions in periodontal health and disease [14] by comparing their levels in oral fluids. The aim was to analyze concomitantly the ionic profile of GCF and saliva in order to explore which of the two types of sample and which combination of chemical elements in these biological matrices could better discriminate between periodontitis and periodontal health.

## 2. Materials and Methods

### 2.1. Study Group

A total of 54 subjects (39 males and 15 females, aged 35–69 years) were consecutively enrolled in this cross-sectional study among those seeking oral health consultation or undergoing supportive periodontal treatment at C.I.R Dental School, University of Turin (Italy) from June 2019 to January 2020. The research protocol was approved by the Institutional Ethics Committee and informed consent was provided by all participants. All procedures complied with the rules of the Helsinki Declaration. The STROBE guidelines were followed to ensure the proper reporting of this study.

Enrolled patients were Caucasians and had at least 15 remaining teeth. They were classified into three groups according to their periodontal conditions based on the clinical and radiographic criteria proposed by the 2017 International World Workshop on the Classification of Periodontal and Peri-implant Diseases and Conditions [19,20] and met the following criteria:Untreated severe periodontitis: subjects suffering from stage III or IV periodontitis, grade B or C. They had radiographic evidence of bone loss extending to the middle or apical third of the root; interdental clinical attachment (CAL) loss ≥ 5 mm, probing depth (PD) ≥ 6 mm and tooth loss due periodontitis. They had not received any periodontal treatment within 1 year before enrollment.Treated periodontitis: patients with a past diagnosis of severe periodontitis who at the end of the active periodontal treatment had a reduced but stable periodontium (no sites with PD > 4 mm or PD = 4 mm with bleeding on probing (BoP), and full-mouth bleeding score (FMBS) < 10%).Periodontally healthy subjects: they had no history of periodontitis, no radiographic evidence of bone loss, no sites with PD > 3 mm, no interdental CAL loss, and FMBS < 10%.

Exclusion criteria for all the groups were: pregnancy, lactation, current smoking, antibiotic and/or anti-inflammatory therapies in the previous 3 months, any systemic condition that could influence the course of periodontal disease (i.e., diabetes mellitus, renal disease, hepatic disease, history of cardiovascular disease, immunological and auto-immune disorders, organ transplantation) and use of medications able to affect the manifestation of periodontal disease (i.e., calcium channel blockers, cyclosporine, phenytoin). Subjects with BMI ≥ 30 kg/m^2^, alcohol consumers, occupationally exposed to metals or using dietary supplements were also excluded.

### 2.2. Periodontal Examination and Oral Biofluids Collection

All participants received a comprehensive periodontal examination, along with whole-mouth intraoral periapical radiographs using a long-cone technique, by one trained and calibrated investigator. The intra-examiner variability was 0.12 mm for PD and 0.15 mm for CAL. The following periodontal measurements were assessed using a manual periodontal probe (PCPUNC-15, Hu-Friedy^®^, Chicago, IL, USA) at six sites of all teeth excluding third molars: presence/absence of bacterial plaque, presence/absence of BoP, PD and CAL. The percentage of bleeding sites/total sites (FMBS) and the percentage of sites harboring plaque/total sites (full-mouth plaque score, FMPS) were calculated for each subject. 

At least 24 h later, after periodontal measurements, unstimulated 3 mL whole saliva samples were obtained between 08:00 am and 10:00 am according to the procedure described by Silwood et al. [21]. Patients were asked to refrain from drinking alcohol in the previous 12 h, from consuming food, sugar drinks and caffeine and from using toothpastes and mouthwashes in the morning of the saliva harvesting in order to avoid alterations in its composition. The subjects were instructed to let the saliva pool in the floor of their mouth and to expectorate into a graduated sterile tube for about 5 min.

In the same session, GCF was collected using the intracrevicular absorption technique [22,23]. Four sites, one for each quadrant, were selected as sampling sites. In the untreated periodontitis group, we selected periodontal pockets with the highest PD along with clinical signs of inflammation and radiographic evidence of alveolar bone loss. In the treated periodontitis group, the samples were taken from diseased sites that after periodontal treatment exhibited PD ≤ 3 mm and absence of inflammation. In the healthy group, the sites chosen for sampling had PD ≤ 3 mm and were BoP negative. After gently drying, supragingival plaque was removed and the area was isolated using cotton rolls to avoid saliva contamination. GCF was collected by inserting a paper strip (PerioPaper Strips, Oraflow Inc., Plainview, NY, USA) for 30 s into the gingival sulcus/pocket until a mild resistance was felt. Samples contaminated with blood were discarded. The volume of collected GCF in each strip was quantitated with a calibrated electric measuring device (Periotron 8000, Oraflow Inc., Plainview, NY, USA) and converted to an actual volume (µL) by reference to the standard curve [23]. Then, the strips of each patient were pooled together in a coded Eppendorf tube. Samples were stored at −80 °C for subsequent assays [24].

### 2.3. Laboratory Analysis

Qualitative and quantitative analysis of mineral content of saliva and GCF samples was carried out by a blinded examiner using sector field inductively coupled plasma mass spectrometry (SF-ICP-MS) and inductively coupled plasma optical emission spectroscopy (ICP-OES). GCF samples were first eluted from the absorbent paper strips by placing them in 1 mL of phosphate buffered saline (PBS). GCF and saliva samples were centrifuged at 3500 rpm for 10 min and an aliquot of 500 µL from the supernatant was diluted at 5 mL with 1% *v/v* sub-boiled nitric acid (HNO_3_) solution [25].

SF-ICP-MS (Thermo Finnigan Element 2) was used to determine elements present in lower concentrations. Mass resolution and isotope selection were optimized for each element to maximize sensitivity and guarantee resolution of spectral interferences. The isotopes ^63^Cu, ^65^Cu, ^66^Zn, ^67^Zn and ^68^Zn were monitored at low resolution (R = 400), while ^7^Li, ^27^Aluminum (Al), ^55^Mn, ^56^Fe, ^57^Fe, ^85^Rubidium (Rb), ^116^Tin (Sn), ^118^Sn and ^120^Sn at medium resolution (R = 4000). Each sample analysis was conducted after a 60 s uptake and stabilization period. Nine (3 runs × 3 passes) and 12 (4 runs × 3 passes) replications for each selected isotope were performed in low and medium resolution, respectively. Between one sample and another one, the nebulizer system was rinsed for 2 min with 2% sub-boiled HNO_3_ to remove carry-over and recondition the sampler cone. The power applied was 1270 W, 1 L/min flow of both auxiliary and nebulizer gasses, whereas plasma gas was fluxed at 16 L/min. The instrument has a glass concentric nebulizer and a Twinnabar (cyclonic) spray chamber. Ca, K, Mg and Na, which are present in higher concentration, were determined by ICP-OES (PerkinElmer, model Optima 7000 DV, Milan, Italy), provided with a Mira-Mist nebulizer and a cyclonic spray chamber. The power applied was 1300 W. Plasma gas was fluxed at 15 L/min while auxiliary and nebulizer gas flows were 0.2 and 0.6 L/min, respectively. The signals were evaluated in triplicate.

Sets of blanks and calibration verification controls were performed at recurrent intervals during the batch sequences for both SF-ICP-MS and ICP-OES analyses. Limits of detection (LODs), corresponding to three times the standard deviation of the reagent blank, were determined by ICP-OES and SF-ICP-MS [26].

### 2.4. Statistical Analysis

Statistical analyses were performed using a statistical program (SPSS, version 25.0, Chicago, IL, USA). The Shapiro–Wilk test and Q-Q normality plots were applied to verify the normal distribution of the continuous variables. The significance of the differences in periodontal parameters and chemical element concentration in GCF and saliva among the experimental groups was determined using the ANOVA or Kruskal–Wallis test as appropriate, followed by pairwise multiple comparison (Tukey test or Dunn test). The statistical significance of correlations between chemical elements in GCF and saliva was determined using the Spearman correlation test.

The multivariate hierarchical cluster analysis (HCA) was applied to discriminate patients according to the mineral content of GCF and saliva samples using the Ward’s method of agglomeration and the squared Euclidean distance as a measure of the similarity between observations. The results were reported in the form of a dendrogram. Variables were standardized. *p* values < 0.05 were considered statistically significant.

## 3. Results

Demographic characteristics and periodontal measurements are summarized in Table 1. Gender distribution and mean age were comparable among the three experimental groups (*p* = 0.758 and *p* = 0.053, respectively). Statistically significant differences emerged when comparing healthy controls with untreated periodontitis patients in terms of mean FMPS, FMBS and PD (all *p* < 0.001), while clinical conditions were comparable between patients with treated periodontitis and periodontally healthy subjects.

### 3.1. SF-ICP–MS- and ICP-OES-Based Ionic Profiling in GCF and Saliva

Table 2 and Table 3 show the concentrations of mineral elements in GCF and saliva. All GCF samples, except one from the healthy control group, were analyzed using the ICP techniques. Out of the 12 examined chemical elements, ICP-OES was able to detect Na, K, Ca and Mg in GCF, while the levels of the other chemical elements (Al, Rb, Li, Fe, Cu, Zn, Mn and Sn) were below the detection threshold of SF-ICP-MS. Although in a previous report we were able to detect other chemical elements in saliva of periodontitis patients [14], here we considered only those detectable in GCF.

In GCF (Table 2) the levels of Na and K were significantly higher in untreated periodontitis compared to both treated periodontitis and clinically healthy conditions (*p* = 0.009 and *p* < 0.001, respectively), while the amount of Mg was found comparable among groups (*p* = 0.589). The levels of Ca were higher in untreated periodontitis, but the difference did not reach statistical significance (*p* = 0.571).

When considering the trend of such chemical elements in saliva of the same subjects (Table 3), only Na levels were found to be significantly higher in untreated periodontitis patients compared to patients with restored healthy periodontium (*p* = 0.027) and healthy controls (*p* = 0.024).

### 3.2. Correlation Analysis

Table 4 and Table 5 summarize the bivariate correlations of Na, K, Ca and Mg in GCF and saliva according to the periodontal health/disease status. In GCF samples (Table 4) Na was positively correlated with K, irrespective of the periodontal conditions. Mg was strongly correlated with Ca in periodontal health on an intact (*r* = 0.723) or reduced periodontium (*r* = 0.673) and with K in untreated periodontitis (*r* = 0.572). Finally, Na levels were associated with Ca, but only in treated periodontitis (*r* = 0.737).

In saliva (Table 5), Na was directly correlated with Mg and Ca in the three experimental groups, and Mg with K only in treated periodontitis subjects (*r* = 0.602) and in the healthy controls (*r* = 0.635).

### 3.3. Cluster Analysis

An exploratory HCA was performed to assess the ability of the levels of Na and K in GCF and saliva to discriminate patients with treated and untreated periodontitis from periodontally healthy subjects. Figure 1 and Figure 2 report the corresponding dendrogram. HCA based on GCF ionic content enabled the division of subjects with untreated periodontitis (with three exceptions) and those with periodontal health status (with one exception) in two different groups as shown in the dendrogram in Figure 1A. It did not distinguish between patients with stable but reduced periodontium after successful active periodontal treatment and healthy controls (Figure 1B).

The same chemical elements in saliva demonstrated a poor clustering ability (Figure 2): untreated periodontitis patients were not effectively discriminated from healthy subjects (Figure 2A) and treated periodontitis were grouped together with healthy controls (Figure 2B).

## 4. Discussion

This study analyzed the trend of chemical elements in both saliva and GCF across different periodontal conditions and compared the ability of these biological matrices to discriminate between healthy and diseased periodontal status.

We targeted the ionomics analysis on 12 trace and major chemical elements based on the promising results of a preliminary study assessing the behavior of 39 inorganic ions in saliva. Considering the lack of data in the literature and hypothesizing very low levels for trace and ultra trace elements in GCF, we used SF-ICP-MS along with ICP-OES for the analysis of major elements. Harrington et al. applied a mineralomic approach using the ICP-OES to determine Ca, Mg, Na, K and ICP-MS for Zn, Cu, Se, Cr, Mn and Fe in human blood and serum [27]. This latter omics technique has proven effective in the analysis of trace element concentration in body fluid samples owing to the low detection limit, down to the pg/L, in combination with the high resolving power [28].

Analysis of blank solutions containing known amounts of the target elements was also carried out to ensure the accuracy of the analytical method. Nevertheless, the trace elements Li, Al, Mn, Fe, Cu, Zn, Sn and Rb were not quantified in GCF, in contrast to the results of a previous report in saliva [14]. For this reason, the analysis was focused only on the major elements Na, K, Ca and Mg in both fluids. A possible explanation is that SF-ICP-MS is not sensitive enough to reach changes in GCF medium considering that we collected a mean GCF volume of 1.3 µL.

The GCF collection method was derived based on previously published protocols [23,29] and samples were pooled from four sites in order to increase the ability to determine the elemental profile of each subject and to serve as a patient-based diagnostic tool for periodontitis [30]. While individual GCF samples reflect the local inflammatory periodontal tissue conditions, pooled samples may characterize broader patient periodontal status.

Interestingly, Na concentrations were altered by active periodontitis in both saliva and GCF, whereas K levels only in GCF. The K ion probably derives from intracellular sources in inflamed gingival tissues and from the degeneration of epithelial and connective tissue within the periodontal pocket [12,31]. Koregol et al. [12] found the level of Na and K higher in GCF of periodontitis compared with gingivitis, but Bang et al. reported opposite results [17]. Increased Na content has been also demonstrated in saliva of chronic and aggressive periodontitis with respect to gingivitis and periodontal health [6,7,8,32]. Na is stored in calcified tissue and thus alveolar bone resorption caused by periodontitis could result in the release of a large quantity of such metal ion into the extracellular compartment and then into GCF and saliva [12]. It is also possible that the inflammatory milieu impairs active Na reabsorption through the pocket epithelium [12,31]. This is supported by the positive correlation between Na imbalance in both GCF and saliva and clinical periodontal indices (BoP, PD and CAL loss) [8,12,16,33]. However, these findings did not agree with those by Bang et al. [17] reporting a negative correlation between bone loss and Na concentration in GCF. Elevation of Na in oral fluids could be also interpreted as a host defense mechanism due to its antimicrobial action [32].

Concerning Mg, its behavior has been negatively related to that of K ions. Aun [7] found enhanced K concentration in oral fluids of periodontitis patients along with Mg deficiency. Mg ions increase the efflux of K from cells via Mg^2+^-sensitive K channels [34] and exert an anti-oxidant activity through the reduction in peroxide radical production. Mg deficiency is also involved in the activation of neutrophils and acute phase proteins and in the modulation of endothelial function [35]. A significant increase in GCF levels was observed along with clinical inflammatory parameters [12,16]. Kasuma [36] reported an inverse relationship between the GCF levels of Mg and periodontal status with the highest amount in healthy conditions and the lowest in mild periodontitis. The present results would seem to support these findings.

Notably, we found a tendency for Ca to elevate in GCF of periodontitis patients, even if the differences with healthy subjects did not reach the statistical significance due to the relative wide range of variation between individuals. In contrast, previous studies suggested that Ca levels were comparable in GCF of periodontitis, gingivitis and healthy groups [12,17] and were not correlated to periodontal clinical measurements [17].

The Ca ion is an essential mediator for intracellular signaling pathways, which during periodontal inflammation are upregulated to stimulate reactive oxygen species production and cytokine expression [37]. It modulates the homeostasis of dental tissues and influences the mineralization of dental plaque and hence calculus formation, which in turn is a predisposing factor for the development of gingivitis that can evolve into periodontitis [38]. To this regard, Ashley reported a positive correlation between the Ca levels in dental plaque and saliva [39]. Elevated concentrations of salivary Ca have been also related to active periodontal involvement and interpreted as the result of the increased alveolar bone loss [6,7,10,40]. Our finding that Ca levels were similar in both GCF and saliva of treated periodontitis and healthy subjects could be attributed to the periodontal infection/inflammation control achieved by means of the active periodontal treatment.

Hence, the concentration of Na, K and Ca in GCF would seem to reflect the clinical status of the periodontal tissues, so that their estimation may be used as a potential diagnostic marker of an active disease status within the periodontal environment. The local increase in metal ions was more prominent in the periodontal region/pocket and was more significant than changes observed in whole saliva. It is also possible that multiple ions act synergically and influence the overall ion homeostasis in oral fluids, but it still remains unknown whether ion networks exert a cluster effect on periodontal health [11].

In this context, the correlation analysis confirmed the different mutual dependences of the targeted ions in the three periodontal groups. In the hierarchical cluster analysis Na and K combination was effective in discriminating untreated periodontitis from the treated counterpart and healthy periodontal conditions in GCF, but not in saliva. Intact and reduced healthy periodontium after active periodontal therapy were clustered into the same group suggesting that oral ion homeostasis can be restored by successful periodontal treatment. In a previous study, we demonstrated the ability of saliva to differentiate periodontitis from healthy periodontal status based on the levels of five metal ions [14].

Considering that chemical elements are involved in the modulation of the inflammatory and immune response, it would be intriguing to explore the association between periodontitis and levels of metal ions, matrix metalloproteinases (MMP) and interleukins (ILs) in the oral fluids. Recent systematic reviews focused on IL-1beta, IL-6 and MMP-8 as promising candidates for early diagnosis of periodontitis [41,42]. These mediators induce bone resorption and connective tissue destruction and have demonstrated a significant positive correlation with periodontal clinical parameters and rate of bone loss. However, due to the large variability in specificity and sensitivity for each individual biomarker, combinations of multiple mediators are required to enhance the diagnostic accuracy. In this regard, ions can act as key environmental signals that modify the behavior of pro-inflammatory mediators. Activities of MMPs seem to vary according to the different metals and their concentrations [43] and K levels alter the immune response of gingival epithelium modifying the expression of IL-6 [44].

This study has some limitations. First of all, its cross-sectional design prevents us from establishing any definitive conclusion on the relationship between periodontitis and ionic imbalance in the oral fluids. However, the similarity of elemental profile in treated periodontitis patients and healthy controls would suggest that elevated levels of metal ions in saliva and GCF are associated to the inflammatory status and the severity of the periodontal breakdown rather than being a risk factor for periodontitis. Certainly, prospective studies would add valuable information in this regard.

It should be also considered that we could not discriminate between bacteria and host cell products. Elements of the metabolism of both bacterial and host origin are released in the GCF and contribute to the final content of whole saliva [23]. In addition, the quantity of collectable GCF can be widely variable among different subjects, above all in periodontal healthy conditions. Moreover, contamination with bacterial plaque, saliva and blood is always possible [45].

Finally, the size of the sample population might have prevented some differences from reaching statistical significance. We applied strict selection criteria in terms of individual lifestyle and medical status. It is known that the content in macro and trace elements may vary with the dietary element intake, smoking habits, use of medications and concurrent systemic diseases [46,47]. These selection criteria might also limit the external generalizability of our findings.

## 5. Conclusions

Based on the present results, we observed a different mineral pattern in GCF and saliva suggesting that biological matrices/tissues from the same patients may have a distinctive ionic profile. This seems to support a tissue specificity of the ionome and merits further investigation.

GCF is a promising biological matrix with higher clustering potential compared with saliva as it could correctly assign a subject to the periodontitis or healthy group based on the combination of only two metal ions. This may be attributed to the fact the GCF profile is more influenced by the quality of the subgingival environment, while the saliva mirrors the whole-mouth inflammatory status as well as the individual’s systemic health conditions. Na and K may represent promising diagnostic biomarkers but considering that we enrolled only stage III periodontitis patients, future research is required to evaluate their predictive ability across the different stages of periodontitis severity.

Our ionomics approach provides some insights between trace elements and periodontitis: cation dyshomeostasis could be employed in the early detection of disease and/or in the monitoring of clinical progression/response to the periodontal treatment. Moreover, it may aid in the identification of novel specific markers for this pathology and may facilitate the development of new drugs and therapeutic strategies.

## Figures and Tables

**Figure 1 biomedicines-10-00687-f001:**
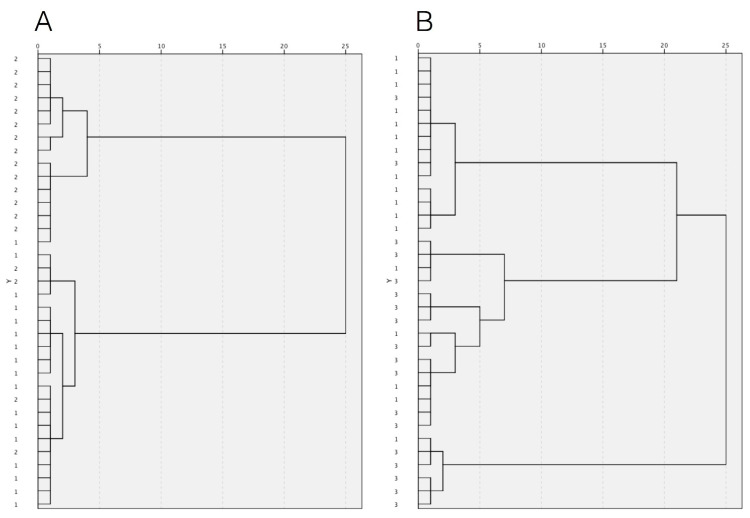
The dendrograms show the results of HCA based on ion distribution in GCF: (**A**) clustering of subjects with untreated severe periodontitis (group 2) and periodontally healthy controls (group 1) and (**B**) clustering of subjects with treated severe periodontitis (group 3) and periodontally healthy controls (group 1).

**Figure 2 biomedicines-10-00687-f002:**
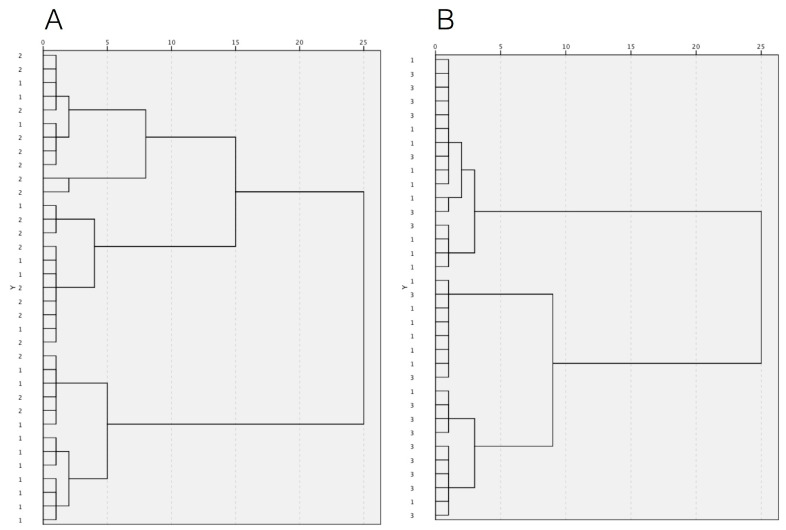
The dendrograms show the results of HCA based on ion distribution in unstimulated whole saliva: (**A**) clustering of subjects with untreated severe periodontitis (group 2) and periodontally healthy controls (group 1); (**B**) clustering of subjects with treated severe periodontitis (group 3) and periodontally healthy controls (group 1).

**Table 1 biomedicines-10-00687-t001:** Demographic and clinical characteristics of the experimental groups.

Parameters	Healthy Controls (*n* = 18)	Untreated Periodontitis (*n* = 18)	Treated Periodontitis (*n* = 18)
Age (years), mean ± SD	46.9 ± 10.5	55.7 ± 10.3	52.7 ± 11.5
Gender (male), (%)	13 (72.2)	14 (77.8)	12 (66.7)
Number of teeth, mean ± SD	28.8 ± 1.8	25.5 ± 4.5	26.0 ± 2.5
FMPS (%), mean ± SD	11.7 ± 2.3	77.1 ± 17.4	14.4 ± 3.5
FMBS (%), mean ± SD	8.2 ± 1.6	67.2 ± 25.5	8.4 ± 1.6
PD (mm), mean ± SD	2.2 ± 0.2	4.2 ± 0.5	2.7 ± 0.4

**Table 2 biomedicines-10-00687-t002:** Distribution of chemical elements in GCF.

Ions	Healthy Controls (*n* = 18)		Untreated Periodontitis (*n* = 18)		Treated Periodontitis (*n* = 18)	
	Mean ± SE	Median (Range)	Mean ± SE	Median (Range)	Mean ± SE	Median (Range)
Na (µg/L)	378.94 ± 10.77	362.01 (327.12–481.26)	517.94 ± 29.58 ^A,B^	515.72 (327.54–762.74)	440.13 ± 23.18	450.85 (325.87–682.78)
Mg (µg/L)	185.54 ± 24.77	169.43 (87.83–453.14)	161.22 ± 17.08	145.32 (98.20–428.66)	175.01 ± 24.56	127.73 (86.27–431.08)
K (µg/L)	54.32 ± 3.05	54.31 (30.91–73.82)	75.18 ± 6.33 ^A,B^	76.41 (34.40–132.03)	44.43 ± 3.49	42.58 (22.11–69.89)
Ca (µg/L)	5410.49 ± 331.87	4810.62 (3714.40–8886.54)	6566.81 ± 421.17	6492.43 (3581.83–10,886.64)	5554.31 ± 219.69	5395.95 (3720.94–6975.00)

Superscript A = *p* < 0.05 compared with healthy group. Superscript B = *p* < 0.05 compared with treated periodontitis group.

**Table 3 biomedicines-10-00687-t003:** Distribution of chemical elements in saliva.

Ions	Healthy Controls (*n* = 18)		Untreated Periodontitis (*n* = 18)		Treated Periodontitis (*n* = 18)	
	Mean ± SE	Median (Range)	Mean ± SE	Median (Range)	Mean ± SE	Median (Range)
Na (mg/L)	174.61 ± 22.73	207.88 (53.10–322.20)	274.81± 23.20 ^A,B^	260.55 (145.90–488.60)	176.31 ± 22.05	182.50 (52.00–377.80)
Mg (mg/L)	8.58 ± 1.04	7.6 (2.38–19.41)	7.81 ± 1.22	6.04 (2.21–20.95)	6.29 ± 0.89	5.25 (1.39–15.82)
K (mg/L)	1092.92 ± 74.23	984.96 (684.60–1758.47)	964.67 ± 82.24	940.08 (382.70–1714.47)	916.96 ± 105.03	839.10 (423.90–2081.87)
Ca (mg/L)	30.77 ± 3.56	31.94 (13.09–66.11)	24.78 ± 3.06	21.05 (9.39–55.10)	26.04 ± 2.82	23.35 (7.09–52.46)

Superscript A = *p* < 0.05 compared with healthy group. Superscript B = *p* < 0.05 compared with treated periodontitis group.

**Table 4 biomedicines-10-00687-t004:** Correlation matrix of mineral elements in GCF according to the periodontal conditions.

	Healthy Controls				Untreated Periodontitis				Treated Periodontitis			
	Na	Mg	K	Ca	Na	Mg	K	Ca	Na	Mg	K	Ca
Na	1.000	0.321	0.520 *	–0.032	1.000	0.316	0.802 **	0.220	1.000	0.342	0.612 **	0.737 **
Mg		1.000	0.348	0.723 **		1.000	0.572 *	0.269		1.000	−0.111	0.673 **
K			1.000	0.292			1.000	0.230			1.000	0.374
Ca				1.000				1.000				1.000

* *p* < 0.05; ** *p* < 0.01.

**Table 5 biomedicines-10-00687-t005:** Correlation matrix of mineral elements in saliva according to the periodontal conditions.

	Healthy Controls				Untreated Periodontitis				Treated Periodontitis			
	Na	Mg	K	Ca	Na	Mg	K	Ca	Na	Mg	K	Ca
Na	1.000	0.681 **	0.620 **	0.520 **	1.000	0.420 *	0.160	0.478 *	1.000	0.772 **	0.583 *	0.608 *
Mg		1.000	0.635 **	0.537 *		1.000	0.220	0.533 *		1.000	0.602 **	0.399
K			1.000	0.350			1.000	0.137			1.000	0.408
Ca				1.000				1.000				1.000

* *p* < 0.05; ** *p* < 0.01.

## Data Availability

The datasets used and analyzed in this study are available from the corresponding author upon request.

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
