# Peer review of "Comparing Ionic Profile of Gingival Crevicular Fluid and Saliva as Distinctive Signature of Severe Periodontitis"

_biomedicines, 2022, doi:10.3390/biomedicines10030687_

Round 1

Reviewer 1 Report

Fine study. I would recommend few more recent references!

Author Response

Dear Editor,

We thank you for the review of our manuscript (biomedicines-1613384) entitled “Comparing ionic profile of gingival crevicular fluid and saliva as distinctive signature of severe periodontitis” by Romano et al. We are pleased that you found our manuscript interesting. We sincerely appreciate the time and the effort you and the Reviewers spent to provide your valuable suggestions, which helped us to improve the quality of the manuscript.

Our responses to the Reviewers’ comments have been reported in a point-by-point manner. Appropriated changes, suggested by the Reviewers, have been introduced in the text (highlighted within the document in yellow).

We hope that you will find our manuscript suitable for publication in its current form.

Best regards,

       Giovanni Nicolao Berta and Coworkers

Reviewer  #1

       Comment 1 to the Authors:  Fine study. I would recommend few more recent references!

Authors’ response/action: Thank you very much for the revision of our manuscript and your suggestion. Unfortunately, the literature on ionic profile of gingival crevicular fluid is very dated. Anyway, we added some recent articles in the paper (see the yellow references).

Reviewer 2 Report

Dear authors,
congratulations for the study, well conducted and well designed.
This study, which analyzes and compares the ionic profile of GCF and saliva of patients with different forms of periodontitis or healthy, shows a difference in their composition and a greater clustering potential of GCF compared to saliva.
Is it possible to indicate the institutional name of the ethics committee and the approval protocol number?
Was the study registered on clinicaltrials.gov or other repositories? Is it possible to provide the registration number?
Has an effect size been calculated?
The clinical / biological significance of the presence of these ions in GCF and saliva has been well explained in the discussions, but what is the clinical relevance of this study? What, if any, are its applications in the clinic?
Thank you

Author Response

Dear Editor,

We thank you for the review of our manuscript (biomedicines-1613384) entitled “Comparing ionic profile of gingival crevicular fluid and saliva as distinctive signature of severe periodontitis” by Romano et al. We are pleased that you found our manuscript interesting. We sincerely appreciate the time and the effort you and the Reviewers spent to provide your valuable suggestions, which helped us to improve the quality of the manuscript.

Our responses to the Reviewers’ comments have been reported in a point-by-point manner. Appropriated changes, suggested by the Reviewers, have been introduced in the text (highlighted within the document in yellow).

We hope that you will find our manuscript suitable for publication in its current form.

Best regards,

       Giovanni Nicolao Berta and Coworkers

Reviewer  #2

Comment 1 to the Authors:  Is it possible to indicate the institutional name of the ethics committee and the approval protocol number? Was the study registered on clinicaltrials.gov or other repositories? Is it possible to provide the registration number?

Authors’ response/action: The requested information is reported at the end of the manuscript before the “References Section” (Institutional Reviewer Board Statement), according to the style of Biomedicines.

Comment 2 to the Authors: Has an effect size been calculated?

Authors’ response/action: Thank you for your useful suggestion. Very few studies are available in literature on the ionic content of the gingival crevicular fluid and none compared the elemental profile of both saliva and gingival crevicular fluid. Considering the lack of data and the explorative design of this research, we did not perform a sample size calculation.

Comment 3 to the Authors: The clinical / biological significance of the presence of these ions in GCF and saliva has been well explained in the discussions, but what is the clinical relevance of this study? What, if any, are its applications in the clinic?

Authors’ response/action: Thank you for your useful observation. We inserted some clinical relevance and possible applications of the study at the end of the “Conclusion Section”.

Reviewer 3 Report

Title: Comparing ionic profile of gingival crevicular fluid and saliva as distinctive signature of severe periodontitis

The topic of this cross-sectional study may be scientifically and clinically interesting for different specialists. Only a few suggestions may be considered as follows:

Keywords

The keywords are appropriate, but 8 are too much. I suggest to remove two of them “metals; saliva”.

Abstract

  • The abstract is too short and not describing the objectives and the rationale of the study.
  • Please, add the acronyms of “plasma-mass spectrometry; inductive coupled plasma optical emission spectroscopy; sodium; potassium; calcium; magnesium” or remove the acronym of “gingival crevicular fluid”.

Introduction

The introduction is well developed and highlights the intentions of the paper. Please, define more in details the aim of the manuscript.

Materials and methods

  • In the “Study Group” section, I suggest to modified “54 adults” with “54 subjects”.
  • In the “Laboratory analysis” section, please add the extended name of “HNO3”.

Results

Table 1: I suggest to remove the description of the table “FMPS: full-mouth plaque score; FMBS: full-mouth bleeding score; PD: probing depth”. You already cited these acronyms in the text. I also suggest to add “mean ± standard deviation” in the table in order to better read the results.

Discussions

  • Discussion section is well developed. However, there are some repetitions of the results and this may be reviewed.
  • Please, use the acronyms of “Sodium; Magnesium; Potassium; Calcium”.

Conclusions

Further discussion should be done in the conclusions to better insert the research in the right perspective.

References

There are too many references. You should reduce them and maybe remark the more important.

Language

The English language may be improved by native speaker.

Author Response

Dear Editor,

We thank you for the review of our manuscript (biomedicines-1613384) entitled “Comparing ionic profile of gingival crevicular fluid and saliva as distinctive signature of severe periodontitis” by Romano et al. We are pleased that you found our manuscript interesting. We sincerely appreciate the time and the effort you and the Reviewers spent to provide your valuable suggestions, which helped us to improve the quality of the manuscript.

Our responses to the Reviewers’ comments have been reported in a point-by-point manner. Appropriated changes, suggested by the Reviewers, have been introduced in the text (highlighted within the document in yellow).

We hope that you will find our manuscript suitable for publication in its current form.

Best regards,

       Giovanni Nicolao Berta and Coworkers

Reviewer  #3

The topic of this cross-sectional study may be scientifically and clinically interesting for different specialists. Only a few suggestions may be considered.

We would like to thank the Reviewer for these very nice comments and his/her time spent in reviewing our paper. The changes are tracked in the revised paper (in yellow color).

Comment 1 to the Authors:  The keywords are appropriate, but 8 are too much. I suggest to remove two of them “metals; saliva”.

Authors’ response/action: We removed “metals” and “saliva” from the list of the keywords as suggested.

Comment 2 to the Authors:  Abstract:

  • The abstract is too short and not describing the objectives and the rationale of the study.
  • Please, add the acronyms of “plasma-mass spectrometry; inductive coupled plasma optical emission spectroscopy; sodium; potassium; calcium; magnesium” or remove the acronym of “gingival crevicular fluid”.

Authors’ response/action: Thank you for useful advice. We improved the abstract to better clarify the objectives and the rationale of the study. We added the acronyms.

Comment 3 to the Authors:  Introduction

The introduction is well developed and highlights the intentions of the paper. Please, define more in details the aim of the manuscript.

Authors’ response/action: Thank you for your comment. We rephrased the sentence to better explain the aim of the study.

Comment 4 to the Authors: Materials and methods

  • In the “Study Group” section, I suggest to modify “54 adults” with “54 subjects”.
  • In the “Laboratory analysis” section, please add the extended name of “HNO3”.

Authors’ response/action: These editing issues have been corrected.

Comment 4 to the Authors: Results

Table 1: I suggest to remove the description of the table “FMPS: full-mouth plaque score; FMBS: full-mouth bleeding score; PD: probing depth”. You already cited these acronyms in the text. I also suggest to add “mean ± standard deviation” in the table in order to better read the results.

      Authors’ response/action: We modified Table 1 as suggested by the Reviewer.

Comment 5 to the Authors: Discussions

  • Discussion section is well developed. However, there are some repetitions of the results and this may be reviewed.
  • Please, use the acronyms of “Sodium; Magnesium; Potassium; Calcium”.

Authors’ response/action: Thank you for your suggestion. We revised the “Discussion Section” and replaced the terms “sodium, magnesium, potassium and calcium” with the corresponding acronyms.

Comment 6 to the Authors: Conclusions

Further discussion should be done in the conclusions to better insert the research in the right perspective.

Authors’ response/action: Thank you for useful advice. We focused the “Conclusion Section” on the values added by the present study.

Comment 7 to the Authors: References

There are too many references. You should reduce them and maybe remark the more important.

Authors’ response/action: We reduced the list of references.

Comment 8 to the Authors: The English language may be improved by native speaker.

      Authors’ response/action: We revised the English language as suggested.

Reviewer 4 Report

Dear Authors,

The article entitled: "Comparing ionic profile of gingival crevicular fluid and saliva as distinctive signature of severe periodontitis" explores  the type and which combination of chemical elements are present in  saliva and gingival crevicular fluid what discriminate between periodontitis and periodontal health.

The article is well-written and exposed clearly the results.

My suggestions are the following:

In MM in the study group description the results on age and gender (lines 91 to 105) should be omitted as they are also presented in Results Table 1. 

In Discussion should be added something about the utility of dosing Ca, Na, K the in clinical practice.

In Conclusions lines 374-377 should be moved to Discussion.

Best regards!

Author Response

Dear Editor,

We thank you for the review of our manuscript (biomedicines-1613384) entitled “Comparing ionic profile of gingival crevicular fluid and saliva as distinctive signature of severe periodontitis” by Romano et al. We are pleased that you found our manuscript interesting. We sincerely appreciate the time and the effort you and the Reviewers spent to provide your valuable suggestions, which helped us to improve the quality of the manuscript.

Our responses to the Reviewers’ comments have been reported in a point-by-point manner. Appropriated changes, suggested by the Reviewers, have been introduced in the text (highlighted within the document in yellow).

We hope that you will find our manuscript suitable for publication in its current form.

Best regards,

       Giovanni Nicolao Berta and Coworkers

Reviewer  #4

Comment 1 to the Authors:  In MM in the study group description the results on age and gender (lines 91 to 105) should be omitted as they are also presented in Results Table 1. 

Authors’ response/action: Thank you for your suggestion. We removed age and gender from the description of the three study groups in the “Materials and Method Section”.

Comment 2 to the Authors: In Discussion should be added something about the utility of dosing Ca, Na, K the in clinical practice.

Authors’ response/action: Thank you for arising this point. We addressed this relevant aspect in the “Conclusion Section” of the manuscript.

Comment 3 to the Authors: In Conclusions lines 374-377 should be moved to Discussion.

Authors’ response/action: Thank you very much for the observation. We followed the suggestion.

Reviewer 5 Report

Dear authors,

The paper written by Romano et al approaches a very interesting topic regarding periodontal diagnosis using oral fluids like GCF and saliva.

I hope that my remarks will be useful in order to increase the quality of the paper.

Line 58 - Please add "Study group" or "research group" instead of just group.

Discussion section. I would recommend to expand this section and also try to include the importance of MMPs and ILs in quantifying periodontal inflammation. I think it will be very useful as comparison to your method of ionic profile.

Conclusions. I would recommend to rephrase this section in such a way that you will emphasise the practical and clinical impact of your results. In the same time you should point out more clearly which biofluid (GCF or saliva) is more relevant for severe periodontitis.

Please receive my best regards!

Author Response

Dear Editor,

We thank you for the review of our manuscript (biomedicines-1613384) entitled “Comparing ionic profile of gingival crevicular fluid and saliva as distinctive signature of severe periodontitis” by Romano et al. We are pleased that you found our manuscript interesting. We sincerely appreciate the time and the effort you and the Reviewers spent to provide your valuable suggestions, which helped us to improve the quality of the manuscript.

Our responses to the Reviewers’ comments have been reported in a point-by-point manner. Appropriated changes, suggested by the Reviewers, have been introduced in the text (highlighted within the document in yellow).

We hope that you will find our manuscript suitable for publication in its current form.

Best regards,

       Giovanni Nicolao Berta and Coworkers

Reviewer  #5

The paper written by Romano et al approaches a very interesting topic regarding periodontal diagnosis using oral fluids like GCF and saliva. I hope that my remarks will be useful in order to increase the quality of the paper.

We would like to thank the Reviewer for his/her thoughtful review and we appreciate the reviewer’s positive feedback. Your insightful comments helped us to strengthen the quality of our paper. The changes are tracked in the revised paper (in yellow color).

Comment 1 to the Authors: 

       Line 58 - Please add "Study group" or "research group" instead of just group.

       Authors’ response/action: Thank you very much for the observation. We followed the suggestion.

Comment 2 to the Authors:  Discussion section. I would recommend to expand this section and also try to include the importance of MMPs and ILs in quantifying periodontal inflammation. I think it will be very useful as comparison to your method of ionic profile.

Authors’ response/action: Thanks to the Reviewer for having raised this issue. We added a paragraph on the utility of IL-1beta, IL-6 and MMP-8 as biomarkers of inflammation in both saliva and GCF. Based on the most recent reviews, we discussed their role as potential candidates for early detection of periodontitis. 

Comment 3 to the Authors:  Conclusions. I would recommend to rephrase this section in such a way that you will emphasise the practical and clinical impact of your results. In the same time you should point out more clearly which biofluid (GCF or saliva) is more relevant for severe periodontitis.

Authors’ response/action: Thank you for your comment. We reorganized the “Conclusion Section” to emphasize the clinical impact of these results.

Round 2

Reviewer 2 Report

All comments have been correctly addressed.

COngratulations obn your article

Reviewer 3 Report

Dear Authors, 

I saw the corrections. 
The manuscript is more clear and precise now.
I accept the manuscript in present form. 

Best regards, 
Ludovica Nucci 

Reviewer 4 Report

Dear Editor,

Thank you for selecting me to review the present article. The data are well presented and the article is good.

The authors followed my recommendations so my opinion for it is to accept in the present form.

Reviewer 5 Report

Dear authors,

You have satisfied all my requirements and I think that the manuscript is suitable for publishing.

Minor piece of advise: for instance at line 360 please use IL1-ß instead of IL1-beta.

Best of luck in your future research!